# Validation of Pharmacological Protocols for Targeted Inhibition of Canalicular MRP2 Activity in Hepatocytes Using [^99m^Tc]mebrofenin Imaging in Rats

**DOI:** 10.3390/pharmaceutics12060486

**Published:** 2020-05-27

**Authors:** Solène Marie, Irene Hernández-Lozano, Louise Breuil, Wadad Saba, Anthony Novell, Jean-Luc Gennisson, Oliver Langer, Charles Truillet, Nicolas Tournier

**Affiliations:** 1Université Paris-Saclay, CEA, Inserm, CNRS, BioMaps, Service Hospitalier Frédéric Joliot, 4 place du Général Leclerc, 91401 Orsay, France; solene.marie@aphp.fr (S.M.); louise.breuil@cea.fr (L.B.); wadad.saba@cea.fr (W.S.); anthony.novell@universite-paris-saclay.fr (A.N.); jean-luc.gennisson@universite-paris-saclay.fr (J.-L.G.); charles.truillet@cea.fr (C.T.); 2Département de Pharmacie Clinique, Faculté de Pharmacie, Université Paris-Saclay, 92296 Châtenay-Malabry, France; 3AP-HP, Université Paris-Saclay, Hôpital Bicêtre, Pharmacie Clinique, 94270 Le Kremlin Bicêtre, France; 4Department of Clinical Pharmacology, Medical University of Vienna, 1090 Vienna, Austria; irene.hernandezlozano@meduniwien.ac.at (I.H.-L.); oliver.langer@ait.ac.at (O.L.); 5Preclinical Molecular Imaging, AIT Austrian Institute of Technology GmbH, 2444 Seibersdorf, Austria; 6Division of Nuclear Medicine, Department of Biomedical Imaging und Image-guided Therapy, Medical University of Vienna, 1090 Vienna, Austria

**Keywords:** drug-induced liver injury, drug metabolism, imaging, liver, membrane transporter, pharmacokinetics

## Abstract

The multidrug resistance-associated protein 2 (MRP2) mediates the biliary excretion of drugs and metabolites. [^99m^Tc]mebrofenin may be employed as a probe for hepatic MRP2 activity because its biliary excretion is predominantly mediated by this transporter. As the liver uptake of [^99m^Tc]mebrofenin depends on organic anion-transporting polypeptide (OATP) activity, a safe protocol for targeted inhibition of hepatic MRP2 is needed to study the intrinsic role of each transporter system. Diltiazem (DTZ) and cyclosporin A (CsA) were first confirmed to be potent MRP2 inhibitors in vitro. Dynamic acquisitions were performed in rats (*n* = 5–6 per group) to assess the kinetics of [^99m^Tc]mebrofenin in the liver, intestine and heart-blood pool after increasing doses of inhibitors. Their impact on hepatic blood flow was assessed using Doppler ultrasound (*n* = 4). DTZ (s.c., 10 mg/kg) and low-dose CsA (i.v., 0.01 mg/kg) selectively decreased the transfer of [^99m^Tc]mebrofenin from the liver to the bile (*k*_3_). Higher doses of DTZ and CsA did not further decrease *k*_3_ but dose-dependently decreased the uptake (*k*_1_) and backflux (*k*_2_) rate constants between blood and liver. High dose of DTZ (i.v., 3 mg/kg) but not CsA (i.v., 5 mg/kg) significantly decreased the blood flow in the portal vein and hepatic artery. Targeted pharmacological inhibition of hepatic MRP2 activity can be achieved in vivo without impacting OATP activity and liver blood flow. Clinical studies are warranted to validate [^99m^Tc]mebrofenin in combination with low-dose CsA as a novel substrate/inhibitor pair to untangle the role of OATP and MRP2 activity in liver diseases.

## 1. Introduction

The channeling of many drugs through hepatocytes is governed by membrane transporters that work together to control hepatic uptake and subsequent biliary elimination [1,2]. Transporters located at the sinusoidal membrane of hepatocytes (blood–liver interface) are responsible for the transfer of compounds between blood and hepatocytes. Transporters located at the canalicular interface (liver–bile interface) mediate the biliary excretion of drugs and/or their metabolites [3] (Figure 1).

Hepatocyte transporters are key determinants for pharmacokinetics (PK) since their altered activity may affect drug exposure, efficacy or safety [4]. Hepatobiliary transporter activity is increasingly considered as a mechanistic determinant for drug-induced liver injury (DILI), which has become a major challenge in drug development [4,5,6]. Many endogenous compounds and xenobiotics are substrates or inhibitors of multidrug resistance-associated protein 2 (MRP2), a major efflux transporter located at the canalicular pole of hepatocytes (Figure 1) [7,8]. MRP2 is the main driving force for bile salt-independent bile flow through canalicular excretion of reduced glutathion [9]. Prolonged depletion or inhibition of MRP2 activity can cause accumulation of harmful bile constituents resulting in cholestasic liver cell damage [9]. In addition, a decrease in MRP2 activity may lead to hepatic accumulation of MRP2-substrate drugs and cause idiosynchratic hepatotoxicity [4]. Selective determination of the individual transport capacity of MRP2 apart from other hepatocyte transporters is needed to address its functional role and regulation during patho-physiological conditions [10].

In vitro hepatocytes or in situ rodent liver perfusion have convincingly demonstrated the importance of canalicular MRP2 activity in controlling the liver-to-bile gradient [11,12]. Dynamic imaging methods using transporter probe substrates are being developed to study liver transporter activity in vivo [13,14]. Ideal imaging probes have to survive hepatic metabolism to avoid the generation of radiolabeled metabolites, which may confound the interpretation of the imaging data [14]. Moreover, sufficient initial liver uptake, which often involves uptake transport, is required to obtain a quantifiable imaging signal in the liver and bile. Radiolabeled compounds that fulfill these criteria are rare. [^99m^Tc]mebrofenin is a readily available radiopharmaceutical used in nuclear medicine for hepatobiliary scintigraphy [15]. [^99m^Tc]mebrofenin undergoes neither metabolism nor conjugation [16,17,18] and has gained interest as a transporter imaging probe because its sinusoidal uptake is highly dependent on organic anion-transporting polypeptide (OATP) influx transporters [17,19,20] and its biliary excretion is predominantly mediated by MRP2 [21,22,23]. In vitro studies have shown that [^99m^Tc]mebrofenin is also transported by the sinusoidal efflux transporter MRP3, which may impact its liver kinetics in vivo [17] (Figure 1). [^99m^Tc]mebrofenin benefits from an extensive hepatic extraction and a fast blood clearance. However, fast clearance from the liver volume and rapid accumulation of radioactivity in the bile makes [^99m^Tc]mebrofenin more suitable to assess biliary function rather than liver extraction [24].

The most effective way to study in vivo the activity of a transporter with a radiolabeled transporter substrate is the use of a specific transporter inhibitor [25,26]. The difference in pharmacokinetic parameters from scans acquired in the presence or absence of a transporter inhibitor unveils transport capacity. An example is the combination of OATP substrate radiotracers (such as [^11^C]dehydropravastatin) with the OATP inhibitor rifampicin (RIF) to study OATP activity in the liver [27]. The possibility of following a similar approach for the imaging of MRP2 activity in the liver using [^99m^Tc]mebrofenin critically depends on the availability of a clinically applicable MRP2 inhibitor which shows negligible interaction with sinusoidal uptake transporters.

There are limited data regarding the impact of transporter inhibitors on the liver kinetics of [^99m^Tc]mebrofenin in vivo. RIF is a potent in vitro inhibitor of OATP [28], MRP2 [8] and MRP3 [29]. In vivo experiments in rats and mice showed that RIF inhibited both the liver uptake and canalicular efflux of [^99m^Tc]mebrofenin to a similar extent [21,30]. Diltiazem (DTZ) is a potent in vitro inhibitor of MRP2 [8], but not of OATP1B1 and OATP1B3 [28]. Cyclosporin (CsA) is a potent in vitro inhibitor of MRP2 [8] and MRP3 [31] but is also known to inhibit OATP [28]. The aim of the present study was to test different doses of DTZ and CsA in rats to come up with an effective and safe imaging protocol, which can be potentially applied in a clinical setting to selectively assess MRP2 activity in the liver with [^99m^Tc]mebrofenin.

## 2. Materials and Methods

### 2.1. Chemicals and Radiochemicals

For in vitro assays, calcein-AM, RIF, DTZ and CsA were obtained from Sigma-Aldrich. For in vivo experiments, commercial drugs were used for RIF (Rifadine^®^, Sanofi, France), DTZ (Tildiem^®^, Sanofi, France) and CsA (Sandimmun^®^, Novartis, France). Commercial kits of mebrofenin (Cholediam^®^) were a gift from Mediam (France). Each kit was labeled with a sodium [^99m^Tc]pertechnetate (Na [^99m^Tc]TcO_4_) eluate (750 MBq/mL) obtained from a sterile [^99^Mo]/[^99m^Tc] generator (Drytec^®^ and Tekcis^®^, GE Healthcare, France) followed by quality control according to the manufacturer’s recommendations.

### 2.2. Animals

[^99m^Tc]mebrofenin imaging experiments were performed in a total of 54 male rats (Wistar, weight: 313 ± 74 g). Doppler ultrasound imaging was performed in an additional 12 male rats (Wistar, 346 ± 67 g). All animal experiments were in accordance with the recommendations of the European Community (2010/63/UE) and the French National Committees (law 2013-118) for the care and use of laboratory animals. The experimental protocol was approved by a local ethics committee for animal use (CETEA) and by the French ministry of agriculture (APAFIS#7466-20 1611 04 1 7049220 v2). Animals were housed in a controlled environment with access to food and water ad libitum.

### 2.3. In Vitro Inhibition Assays

We used an in vitro assay to compare the MRP2-inhibitory potency of RIF, DTZ and CsA. Stably transfected MDCK-II-MRP2 cells overexpressing human MRP2 were a kind gift from A. Schinkel (National Cancer Institute, The Netherlands). Cells were cultured in a controlled atmosphere at 37 °C, 5% CO2. The culture medium was composed of DMEM Glutamax (Dulbecco’s Modified Eagle Medium, 4.5 g/L d-glucose, pyruvate) supplemented with 10% fetal bovine serum and 1% antibiotics (penicillin and streptomycin 5000 U/mL). MRP2-mediated transport was assessed using the calcein-AM assay [32]. Cells were seeded in 24-well plates (30,000 cells per well) in 500 μL of culture medium. Three days later the medium was removed and replaced by the tested inhibitor (RIF, DTZ and CsA) at selected concentrations ranging from 0 (solvent, 1% DMSO) to 1 mM in 200 μL of incubation buffer (10% HBSS (Hanks’ Balanced Salt Solution) + CaCl_2_ + MgCl_2_, 1 mM pyruvate, 10 mM Hepes, 1 μM calcein-AM). After 30 min at 37 °C, the incubation buffer was removed and cell monolayers were rapidly washed with 300 μL of ice-cold buffer (Dulbecco’s phosphate buffered saline + CaCl_2_ + MgCl_2_). Cells were then lysed by 10 min of incubation with 500 μL of NaOH (10 mM). A total of 200 μL was collected from each well (*n* = 4 wells per condition) and intracellular fluorescence was determined using a Mithras^®^ detector (Berthold, France, wavelength excitation 485 nm/detection 535 nm). For each inhibitor, uptake data were normalized to describe the extent of inhibition from 0 to maximal (100%) inhibition. Nonlinear regression was performed to estimate the 50% inhibitory concentration (IC_50_) of each inhibitor (GraphPad Prism software, version 8.4, San Diego, CA, USA).

### 2.4. [^99m^Tc]mebrofenin Scintigraphy

Rats were anesthetized with isoflurane (3.5% and 1.5–2% in oxygen for induction and maintenance, respectively). [^99m^Tc]mebrofenin scintigraphy was performed using a clinical SPECT-CT camera (Symbia^®^, Siemens, Knoxville, TN, USA) with a Low Energy High Resolution (LEHR) collimator. In each session, 3 rats were placed in a row on the scanner bed and catheters were inserted in the caudal vein. Experiments were performed for control conditions or after injection of the different transporter inhibitors. RIF (40 mg/kg intravenously, i.v.) was injected immediately before [^99m^Tc]mebrofenin as previously described [33]. The PK of subcutaneously (s.c.) injected DTZ (20 mg/kg) has been reported in rats with plasma concentration > 10 µg/mL (25 µM) obtained 60 min after injection [34], consistent with the potency of DTZ to inhibit MRP2, assessed in vitro in the present study. A dose range of DTZ (10, 20 or 40 mg/kg s.c.) was therefore administered 60 min before [^99m^Tc]mebrofenin. CsA was injected i.v. 5 min before [^99m^Tc]mebrofenin, starting from a high clinical dose which was further decreased (5, 1, 0.5, 0.1, 0.01 mg/kg, i.v.). The number of rats used for each condition is reported in Table 1. Rats were injected with [^99m^Tc]mebrofenin (39 ± 4 MBq, i.v.) followed by dynamic planar acquisitions. Acquisitions were terminated by a X-ray CT scan. Dynamic images were reconstructed as 20, 10, 20, and 4 frames with a duration of 15, 30, 60 and 120 s respectively, to provide time-activity curves (TACs) in selected organs.

### 2.5. Analysis of [^99m^Tc]mebrofenin Scintigraphy Data

Images were analyzed with PMOD^®^ software (version 3.9, PMOD Technologies LLC). Regions of interest (ROI) were manually drawn on the liver, intestine, heart (= image-derived blood curve) and urinary bladder. Corresponding TACs were generated by plotting the mean radioactivity amount in each ROI normalized to the injected dose in each animal (cps/MBq) versus time. For descriptive pharmacokinetic analysis, the maximum total amount (X_max_, cps/MBq), the time to reach X_max_ (T_max_, min) and the area under the TAC from 0 to 30 min (AUC_0–30 min_) were determined for each ROI. The distribution of [^99m^Tc]mebrofenin from blood to the liver was assessed by determining the liver to blood AUC ratio (AUCR) from 0 to 3 min (${\mathrm{AUCR}}_{\mathrm{liver}/\mathrm{blood}}=\frac{{\mathrm{AUC}}_{\mathrm{liver},0–3\min}}{{\mathrm{AUC}}_{\mathrm{blood},0–3\min}}$). The distribution of [^99m^Tc]mebrofenin from the liver to the intestine was assessed by calculating AUCR_intestine/liver_ from 10 to 25 min (${\mathrm{AUCR}}_{\mathrm{intestine}/\mathrm{liver}}=\frac{{\mathrm{AUC}}_{\mathrm{intestine},10–25\min}}{{\mathrm{AUC}}_{\mathrm{liver},10–25\min}}$). The urinary clearance (CL_urine_) of [^99m^Tc]mebrofenin was calculated using the equation ${\mathrm{CL}}_{\mathrm{urine}}=\frac{X_{\mathrm{bladder},30\min}}{{\mathrm{AUC}}_{\mathrm{blood},0–30\min}}$) where X_bladder_,_30 min_ represents the amount of [^99m^Tc]mebrofenin in the bladder at 30 min after the tracer injection.

### 2.6. Pharmacokinetic Modeling of [^99m^Tc]mebrofenin Hepatobiliary Distribution and Elimination

The uptake clearance of [^99m^Tc]mebrofenin from blood to the liver (CL_uptake_) was estimated using the integration plot method [33,35] from 0 to 3 min (linear part) after [^99m^Tc]mebrofenin injection using the following equation: $\frac{X_{liver,t}}{C_{blood,t}}=CL_{uptake}\times \left(\frac{AUC_{blood,0–t}}{C_{blood,t}}\right)+V_{E}$. X_liver,t_ represents the total amount of [^99m^Tc]mebrofenin in the liver at the time t, C_blood,t_ represents the concentration of [^99m^Tc]mebrofenin in the blood at time t (derived from the heart ROI). C_blood,t_ was calculated as the amount of [^99m^Tc]mebrofenin in the blood (X_blood,t_) divided by the mean volume of the heart ROI (V_heart_), which was measured on the X-ray CT scan (V_heart_ = 2.55 ± 0.36 mL). AUC_blood,0–t_ represents the area under the blood TAC from time 0 to t. CL_uptake_ corresponds to the slope of the linear regression plot. V_E_ represents the initial distribution volume in the liver at time 0. The transfer of [^99m^Tc]mebrofenin from liver to bile could not be estimated using integration plot analysis as plots were not linear at any time point of the kinetics.

A previously developed four-compartment model [36] was implemented in order to estimate the rate constants that describe the transfer of [^99m^Tc]mebrofenin between blood and hepatocytes (k_1_, and k_2_, min^−1^), from hepatocytes to intrahepatic bile ducts (k_3_, min^−1^), and from the intrahepatic bile ducts to the intestine (k_5_, min^−1^) (Appendix A). The processes defined by k_1_, k_2_, and k_3_ may involve active transport through the basolateral or canalicular membranes of hepatocytes. The liver volume of interest defined in the model consists of well-mixed compartments of blood, hepatocytes and intrahepatic bile ducts. The additional compartment represents the extrahepatic bile ducts and intestines. Compartment volumes are estimated by hepatic anatomical characteristics and physiological processes as previously described [36,37]. In addition, the model includes a flow-weighted dual blood input function that accounts for the tracer delivery to the liver via both the hepatic artery and the portal vein. The radioactivity concentration in the hepatic artery corresponded to the image-derived TACs from the ROI in the heart-blood pool divided by V_heart_, while the concentration in the portal vein was mathematically estimated from the arterial blood concentration and the liver and intestine TACs during the modeling process [36]. The final dual-input concentration was calculated as the flow-weighted input concentrations of hepatic artery and portal vein using a hepatic arterial flow fraction of 0.17 [38].

### 2.7. Assessment of Liver Perfusion with Doppler Ultrasound

The impact of inhibitors on the hepatic blood flow was assessed using Doppler ultrasound imaging (US). Flow measurements were performed in three hepatic vessels (portal vein, hepatic vein and hepatic artery). Rats were anesthetized with isoflurane (3.5% and 1.5–2% in oxygen for induction and maintenance, respectively) and Doppler US imaging was performed by using an ultrafast ultrasound device (Aixplorer Ultimate^®^, V12.3, Supersonic Imagine). The US probe (SLH20-6^®^, Supersonic Imagine) was positioned on the shaved abdomen of each rat. Flow in the three vessels was measured before (= baseline) and 5 min after injection of maximal dose of the tested inhibitors to allow for paired measurements, without moving the US probe, over a limited time period. Therefore, all three inhibitors, including DTZ, had to be injected i.v. to enable rapid detection of drug-induced changes in blood flow within a single session. RIF was administered 40 mg/kg i.v. (*n* = 4) and CsA was administered 5 mg/kg i.v. (*n* = 4). For DTZ, it was not technically possible to match the conditions used for [^99m^Tc]mebrofenin scintigraphy and wait for 60 min after s.c. administration. Intravenous injection of 40 mg/kg dose of DTZ was lethal (*n* = 3). DTZ was therefore administered at the maximal i.v. dose reported in the literature in rats (3 mg/kg i.v.; *n* = 4) [39]. Blood flow in each vessel was estimated from the flow speed measured in Doppler, normalized to the size of each vessel.

### 2.8. Statistical Analysis

[^99m^Tc]mebrofenin scintigraphy data are reported as mean ± standard deviation (SD). Differences between multiple groups were analyzed by one-way ANOVA followed by a Bonferroni multiple-comparison test and between two groups (Doppler US measurements) with a paired *t*-test. The level of statistical significance was set to a *p* value of less than 0.05.

## 3. Results

### 3.1. In Vitro Inhibition Assays

The calcein-AM assay was performed to compare the MRP2 inhibitory potency of selected compounds in vitro (Figure 2). Full MRP2 inhibition was achieved only using high concentrations of RIF (>500 µM). The IC_50_ of RIF was 210 µM (95% confidence interval: 175–254 µM). The IC_50_ for DTZ and CsA was 4.4 µM (95% confidence interval: 1.7–10.4 µM) and 1.7 µM (95% confidence interval: 1.0–2.7 µM), respectively, suggesting a higher MRP2-inhibition potency. CsA appeared to be the most potent MRP2 inhibitor with the lowest IC_50_. Concentrations of CsA > 30 µM could not be tested due to cell toxicity.

### 3.2. Impact of Inhibitors on the Kinetics of [^99m^Tc]mebrofenin In Vivo

[^99m^Tc]mebrofenin kinetics were first measured after treatment with the well-characterized OATP/MRP2 inhibitor RIF. Figure 3 shows representative planar images obtained in a control rat and a RIF-treated rat. Under control conditions, [^99m^Tc]mebrofenin was rapidly transferred to the liver within seconds and then gradually moved to the intestine. The liver uptake of [^99m^Tc]mebrofenin was markedly reduced by RIF with less radioactivity in the intestine at later time points.

Figure 4 shows the mean TACs obtained for each tested condition. Descriptive pharmacokinetic data are reported in Table 1. Unlike RIF, DTZ and CsA increased the liver AUC_0–30 min_ (with a maximum of +51% for 1 mg/kg CsA) while decreasing AUC_0–30 min_ in the intestine (from −23% to −47% with 0.01 and 0.1 mg/kg CsA, respectively). X_max,liver_ was significantly increased by the lowest dose of CsA (0.01 mg/kg) only and T_max,liver_ was significantly delayed by CsA in a dose-dependent manner (+4.8 min with CsA 5 mg/kg). AUC_blood_ showed a pronounced dose-dependent increase after CsA, suggesting lower plasma clearance. Radioactivity in the intestine was significantly decreased in all tested conditions except for the lowest dose of CsA (0.01 mg/kg, Table 1).

The liver uptake of [^99m^Tc]mebrofenin was estimated by AUCR_liver/blood_ from 0 to 3 min (early uptake phase, Figure 5). Compared to the control group (AUCR_liver/blood_ = 2.95 ± 0.47) there was a dose-dependent decrease in liver uptake with increasing doses of DTZ and CsA. The highest doses of DTZ and CsA induced an approximately 3-fold decrease in AUCR_liver/blood_, reaching similar levels as in RIF-treated animals. The lowest doses of DTZ and CsA (10 mg/kg and 0.01 mg/kg, respectively) did not affect the initial liver uptake of [^99m^Tc]mebrofenin. Biliary excretion was estimated by calculating AUCR_intestine/liver_. A time-frame from 10 to 25 min was selected for this analysis, which corresponds to the elimination phase observed in the TACs under all tested conditions. AUCR_lintestine/liver_ was significantly decreased in all tested conditions compared with the control group (Figure 5). Differences between tested conditions were not significant. Nevertheless, CsA appeared to be more potent than DTZ in decreasing AUCR_intestine/liver_.

The urinary bladder could be observed on the [^99m^Tc]mebrofenin images with higher signals in animals which received the highest doses of the inhibitors. However, CL_urine_ values were not significantly different among all tested conditions (Figure 5).

### 3.3. Pharmacokinetic Modeling

The liver uptake of [^99m^Tc]mebrofenin was also estimated with integration plot analysis [33,35] (Appendix A). Compared to the control group (CL_uptake_ = 3.38 ± 0.49 mL·min^−1^), the potent OATP inhibitor RIF significantly decreased the liver uptake (CL_uptake_ = 0.81 ± 0.08 mL·min^−1^) (Figure 5). A dose-dependent decrease in CL_uptake_ was observed with increasing doses of DTZ and CsA. The maximal decrease was observed with 5 mg/kg CsA, reaching similar levels as in RIF-treated animals (CL_uptake_ = 0.81 ± 0.10 mL·min^−1^). The highest dose of DTZ only partially reduced CL_uptake_ (1.85 ± 0.60 mL·min^−1^ for 40 mg/kg DTZ). The lowest dose of DTZ (10 mg/kg) and CsA (0.01 mg/kg) did not decrease CL_uptake_, suggesting a negligible impact on the liver extraction of [^99m^Tc]mebrofenin. The initial distribution volume in the liver (V_E_) was similar across all tested conditions (V_E_ = 1.64 ± 0.38 mL) (Appendix A).

We used a previously developed four-compartment PK model which provided good fits (Appendix A) for both the liver and the intestine ROI. Precision of the obtained parameters was acceptable (low percentage coefficient of variation, %CV) in almost all the analyzed subjects (Appendix A). Compartmental kinetic modeling showed that the transfer rate constant of [^99m^Tc]mebrofenin from blood into hepatocytes *k*_1_ (Appendix A) was significantly decreased by RIF (*k*_1_ = 0.89 ± 0.09 min^−1^) and, to a lesser extent, by the highest doses of DTZ (*k*_1_ = 2.76 ± 0.89 min^−1^) and CsA (*k*_1_ = 2.82 ± 0.86 min^−1^) as compared with the control group (*k*_1_ = 7.37 ± 1.62 min^−1^) (Figure 6). A dose-dependent decrease in *k*_1_ was observed with increasing doses of DTZ and CsA, with no effect observed for 10 mg/kg DTZ and 0.01 mg/kg CsA. The backflux of [^99m^Tc]mebrofenin from hepatocytes into blood *k_2_* (Appendix A) was dose-dependently decreased by DTZ and CsA (Figure 6). *k_2_* was not decreased by the lowest dose of DTZ and CsA. The transfer of [^99m^Tc]mebrofenin from the hepatocyte into the intrahepatic bile ducts *k_3_* (Appendix A), which reflected MRP2-mediated excretion, was significantly decreased in all tested groups to a similar extent (Figure 6) except for animals receiving 5 mg/kg CsA, for whom a high variability in *k*_3_ was observed (*k_3_* = 0.15 ± 0.08 min^−1^). Statistically significant differences remained the same for *k*_2_ and *k*_5_ after excluding outlier data. However, the significance is lost in some of the groups when removing the outlier value for *k*_3_ (0.29 min^−1^) in the control group. In this situation, only the 20 mg/kg DTZ, and all the CsA groups except the 5 mg/kg group show a significant decrease in the *k*_3_ value as compared to the control group. No differences between groups could be observed in the *k*_5_ parameter, which describes the transfer of [^99m^Tc]mebrofenin from the intrahepatic bile ducts into the intestine (Figure 6).

### 3.4. Assessment of Liver Perfusion with Doppler Ultrasound

Individual blood flow values measured in three hepatic vessels in rats before and after the i.v. injection of each inhibitor (40 mg/kg RIF, 3 mg/kg DTZ, 5 mg/kg CsA) are shown in Figure 7. Higher doses of DTZ than 3 mg/kg could not be tested due to poor tolerance. None of the tested inhibitors affected the hepatic vein blood flow. A significant decrease in the portal vein blood flow was observed after injection of RIF and DTZ. Moreover, a significant decrease in the hepatic artery blood flow was observed after injection of DTZ. CsA did not affect the blood flow in any hepatic vessel (Figure 7).

## 4. Discussion

Hepatocyte transporters play a key role in the hepatobiliary clearance of many endogenous and exogeneous compounds. Efforts are being made to develop imaging protocols for selective determination of the respective importance of each transporter system in vivo [40]. Several small-molecule probes for Positron Emission Tomography (PET) have been proposed to study OATP transporter activity and to assess the interplay between sinusoidal uptake and canalicular efflux in vivo [13,19,40]. However, imaging protocols for selective determination of MRP2 activity at the canalicular interface are currently not available. The availability of such an imaging protocol would be of high interest to predict MRP2-mediated DILI in drug development [41] and to study the effect of genetic polymorphisms or liver disease on hepatic MRP2 activity [42].

In vitro, [^99m^Tc]mebrofenin is predominantly transported by sinusoidal OATP1B1 and OATP1B3 and by the efflux transporters MRP2 and MRP3 [17,19,20,22] (Figure 1). In vivo studies using either OATP- or MRP2-deficient mice provided evidence for the transport of [^99m^Tc]mebrofenin by these transporters [30]. RIF is recommended as a prototypical OATP transporter inhibitor to investigate OATP-mediated drug–drug interactions in clinical pharmacokinetic studies [5,43]. RIF has also been used to characterize different OATP-imaging probes in animals and humans [19]. However, our rat data confirmed previous results obtained in mice showing that RIF inhibited both the OATP-mediated uptake and the MRP2-mediated biliary excretion of [^99m^Tc]mebrofenin [30]. Neyt and colleagues have tested the impact of RIF on the liver kinetics of [^99m^Tc]mebrofenin in mice [30]. The lowest tested dose (12.5 mg/kg i.v. followed by 3.12 mg/kg intraperitoneal) did not impact the liver uptake of [^99m^Tc]mebrofenin, but only moderate inhibition of biliary excretion was observed (−25% in AUCR_bile/liver_). Higher doses of RIF inhibited both the OATP-mediated uptake (AUCR_liver/blood_) and the biliary excretion (−94% in AUCR_bile/liver_). Of note, liver kinetics of [^99m^Tc]mebrofenin in RIF-treated mice did not resemble those obtained in MRP2 deficient mice [30], suggesting that RIF inhibits liver OATPs and MRP2 to a similar extent. DTZ is a potent inhibitor in vitro of MRP2 [8], but not of OATP1B1 and OATP1B3 [28]. CsA is a potent MRP2 inhibitor [8] but is also known to inhibit OATPs in vitro [28]. We hypothesized that CsA may show substantially different potency to inhibit MRP2 versus OATP in vivo [21,28]. Our in vitro data (Figure 2) showed that the potency of DTZ and CsA to inhibit MRP2 was much higher than that of RIF.

Both DTZ and CsA induced a dose-dependent decrease in the liver uptake of [^99m^Tc]mebrofenin. However, the lowest tested doses of DTZ (10 mg/kg) and CsA (0.01 mg/kg) did not impact the liver uptake of [^99m^Tc]mebrofenin, estimated as AUCR_liver/blood_, CL_uptake_ or *k*_1_, suggesting that these doses did not affect liver OATP activity in vivo. All tested doses of DTZ and CsA inhibited biliary excretion (estimated as AUCR_intestine/liver_ or *k*_3_) to a similar extent, although DTZ appeared slightly less efficient than RIF and CsA.

DTZ is a calcium channel blocker and we hypothesized that the decrease in the liver uptake of [^99m^Tc]mebrofenin induced by high-dose DTZ may reflect its impact on hepatic blood flow [36]. For technical reasons, it was not possible to test the impact of DTZ 40 mg/kg s.c. on liver perfusion. However, DTZ (3 mg/kg i.v.) was shown to induce a modest but significant decrease in blood flow in both the portal vein and hepatic artery (Figure 7). In contrast, the highest dose of CsA (5 mg/kg i.v.) did not impact the blood flow in any hepatic vessel, suggesting that the effects of CsA on hepatic [^99m^Tc]mebrofenin kinetics reflected changes in transporter activity rather than changes in liver perfusion. We showed that high-dose RIF significantly reduced the blood flow in the portal vein, which is not likely to occur at the doses used in clinical studies (approximately 9 mg/kg) [44,45]. However, it cannot be excluded that changes in portal perfusion induced by high-dose RIF may have influenced to some extent the liver kinetics of [^99m^Tc]mebrofenin in our experiments.

Different clinical imaging modalities have been used to study liver transporter activity in vivo, including magnetic resonance imaging, PET and gamma-scintigraphy [19]. In contrast to PET radioligands, [^99m^Tc]mebrofenin is readily available at most nuclear medicine departments. [^99m^Tc]mebrofenin imaging protocols typically involve dynamic planar acquisition for semi-quantitative estimation of the transfer of radioactivity between tissues [46]. We mimicked these clinical protocols in our set-up, which enabled us to perform compartmental pharmacokinetic modeling with our data. Results obtained for inhibitor-induced changes in the influx and efflux rate constants (*k*_1_ and *k*_3_) were consistent with results obtained for AUCR or CL_uptake_. However, compartmental modeling provided additional information suggesting that the backflux (*k*_2_) from the liver into the blood may influence the liver kinetics of [^99m^Tc]mebrofenin. This could be due to transport of [^99m^Tc]mebrofenin by the sinusoidal efflux transporter MRP3 [17] (Figure 1), consistent with the in vitro potency of RIF and CsA to inhibit MRP3 [29,31], although the impact of DTZ on MRP3 has not been reported to our knowledge. Inhibitors dose-dependently decreased *k*_2_, which suggests that the effects of high-dose RIF, DTZ and CsA may not only be restricted to OATP and MRP2 inhibition [8].

Dynamic in vivo data using a metabolically stable probe offers the possibility to investigate the pharmacokinetic impact of targeted MRP2 inhibition [47]. Interestingly, selective inhibition of the biliary excretion did not affect plasma exposure while increasing liver exposure (Table 1). This observation is consistent with the kinetics of [^99m^Tc]mebrofenin obtained in MRP2-deficient mice [30]. This suggests that MRP2 activity is not a rate-limiting factor for the plasma clearance of this prototypical OATP/MRP2 substrate. MRP2 inhibition may thus drastically affect the liver exposure of MRP2 substrates with negligible impact on plasma kinetics, in agreement with the extended clearance concept [48]. Compared with non-treated rats (AUCR_intestine/liver_ = 1.36 ± 0.20), the decrease in biliary excretion obtained using low-dose CsA (AUCR_intestine/liver_ = 0.67 ± 0.19) was nearly maximal because higher doses and other tested MRP2 inhibitors did not further decrease it. It is noteworthy that the baseline biliary excretion of [^99m^Tc]mebrofenin reported in mice was much higher (calculated AUCR_bile/liver_ ~28) [30]. The impact of MRP2-deficiency was consistently much more pronounced, with a ~10-fold decrease in AUCR_bile/liver_ [30]. This observation suggests a species difference in the importance of MRP2 in controlling the biliary excretion of [^99m^Tc]mebrofenin between rats and mice. In rats, tested inhibitors did not influence the urinary clearance of [^99m^Tc]mebrofenin, thus supporting the assumption that urinary elimination of [^99m^Tc]mebrofenin does not involve carrier-mediated transport [17].

Clinical validation of in vivo inhibitors of membrane transporters is a cornerstone for translational pharmacokinetics [49,50,51]. In this framework, imaging techniques are useful to address the specificity and efficacy of compounds to inhibit drug transporters at certain biological interfaces. DTZ doses used to achieve selective MRP2 inhibition in the liver (10 mg/kg s.c.) were much higher than those clinically used in humans (0.3 mg/kg i.v.) and may additionally alter hepatic blood flow. In contrast, low-dose CsA (0.01 mg/kg), validated in this work, represents a much lower dose than clinical immunosuppressant doses of CsA (up to 15 mg/kg/day) and offers a readily available, simple and safe strategy to inhibit the MRP2-mediated biliary efflux of [^99m^Tc]mebrofenin. Thus, low-dose CsA may be useful both for a selective determination of MRP2 activity and for an improved measurement of OATP activity. Low-dose CsA therefore enriches the toolbox of available inhibitors for pharmacokinetic studies [41].

From a nuclear medicine perspective, our data suggest that patients undergoing treatment with CsA may show substantial inhibition of the biliary excretion of [^99m^Tc]mebrofenin, which may confound the estimation of biliary transporter activity for diagnostic purposes. On the other hand, low-dose CsA may be useful to enhance and prolong the [^99m^Tc]mebrofenin signal in the liver and preferentially study the hepatic uptake activity rather than biliary excretion as a clinical index of liver function in patients.

## 5. Conclusions

This work shows for the first time that targeted inhibition of MRP2 activity with negligible impact on OATP activity in hepatocytes can be achieved in vivo. [^99m^Tc]mebrofenin in combination with low-dose CsA provides a novel substrate/inhibitor pair to discriminate the canalicular transport capacity from the OATP-mediated uptake in vivo. Clinical studies are warranted to validate this protocol in humans and untangle the intrinsic role of OATP and MRP2 activity in different patho-physiological states and address consequences for PK and liver exposure.

## Figures and Tables

**Figure 1 pharmaceutics-12-00486-f001:**
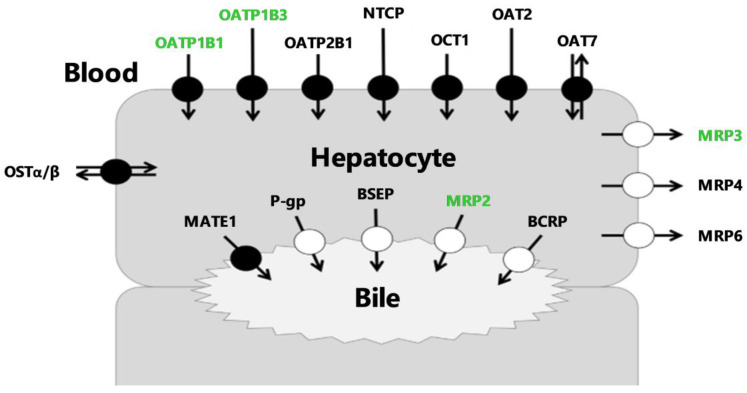
Membrane transporters expressed in hepatocytes. Transporters indicated in green have been described in the literature to transport [^99m^Tc]mebrofenin. BCRP = Breast Cancer Resistance Protein, BSEP = Bile-Salt Export Pump, MATE1 = Multidrug And Toxin Extrusion Protein 1, MRP = Multidrug Resistance-Associated Protein, NTCP = Na^+^-taurocholate-cotransporting polypeptide, OAT = Organic Anion Transporter, OATP = Organic Anion-Transporting Polypeptide, OCT = Organic Cation Transporter, OST = Organic Solute Transporter, P-gp = P-glycoprotein.

**Figure 2 pharmaceutics-12-00486-f002:**
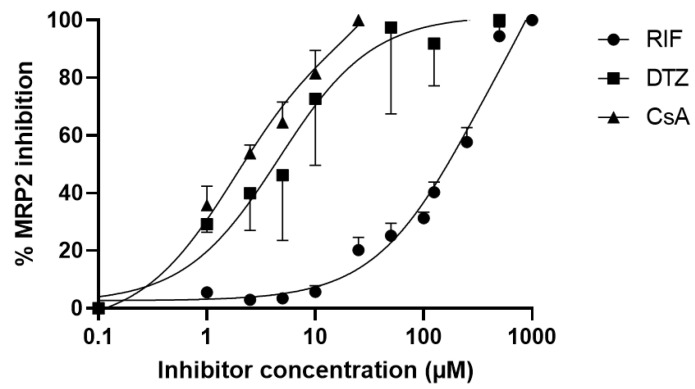
In vitro inhibition of MRP2 activity by rifampicin (RIF), diltiazem (DTZ) and cyclosporin (CsA). MRP2 activity was assessed with the calcein-AM efflux assay using MDCK-II-MRP2 cells incubated with increasing concentrations of tested inhibitors. Data are mean ± S.D (*n* = 4) and lines show the nonlinear regression fit.

**Figure 3 pharmaceutics-12-00486-f003:**
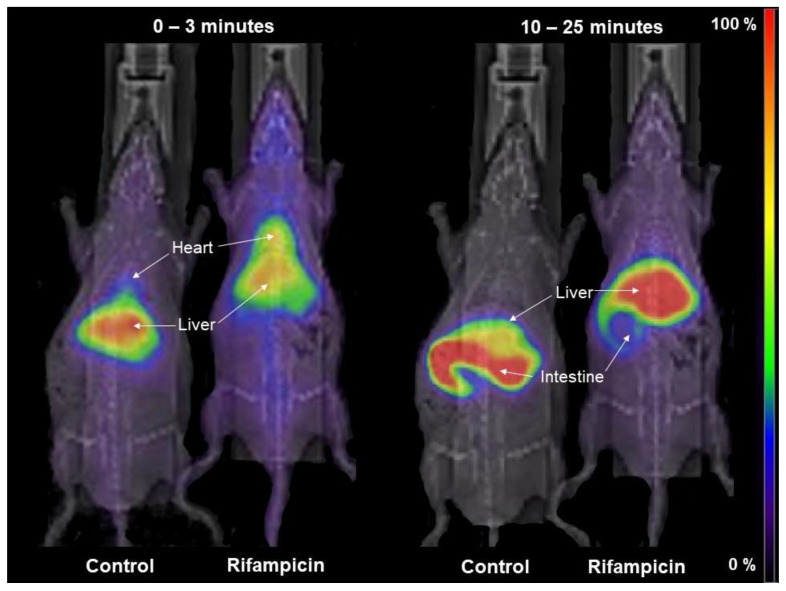
Representative planar scintigraphy images of the distribution of [^99m^Tc]mebrofenin in a control and RIF-treated rat (40 mg/kg i.v.). Shown are summed frames from 0 to 3 min (to depict hepatic uptake) and summed frames from 10 to 25 min (to depict biliary excretion).

**Figure 4 pharmaceutics-12-00486-f004:**
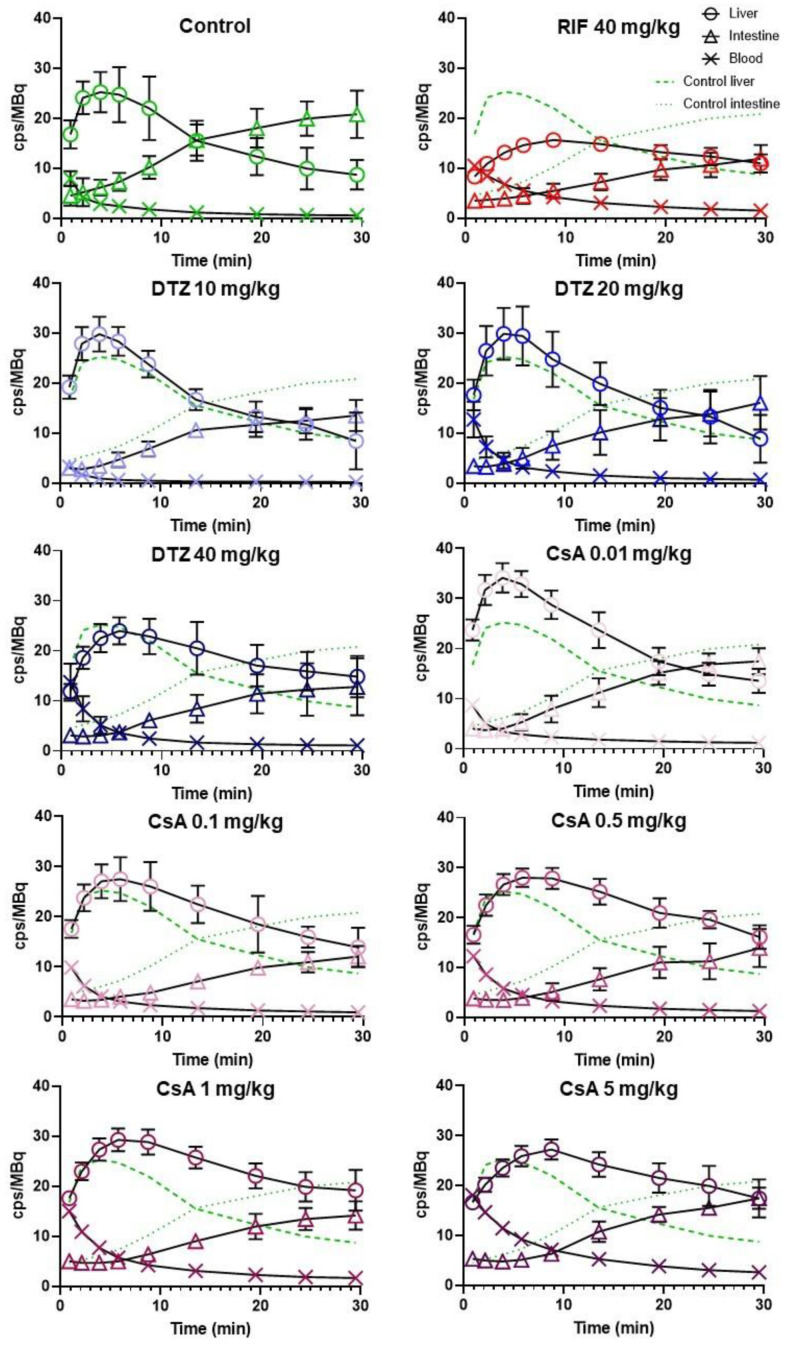
Mean (± SD) time-activity curves in the liver, intestine and blood for control animals and animals treated by rifampicin (RIF), diltiazem (DTZ) or cyclosporin A (CsA) with *n* = 5 for control, RIF, DTZ 10 mg/kg, DTZ 20 mg/kg, DTZ 40 mg/kg, CsA 0.1 mg/kg groups and *n* = 6 for CsA 0.01 mg/kg, CsA 0.5 mg/kg, CsA 1 mg/kg, CsA 5 mg/kg groups. Activity is expressed as the number of counts per second (cps) normalized to the injected dose (MBq).

**Figure 5 pharmaceutics-12-00486-f005:**
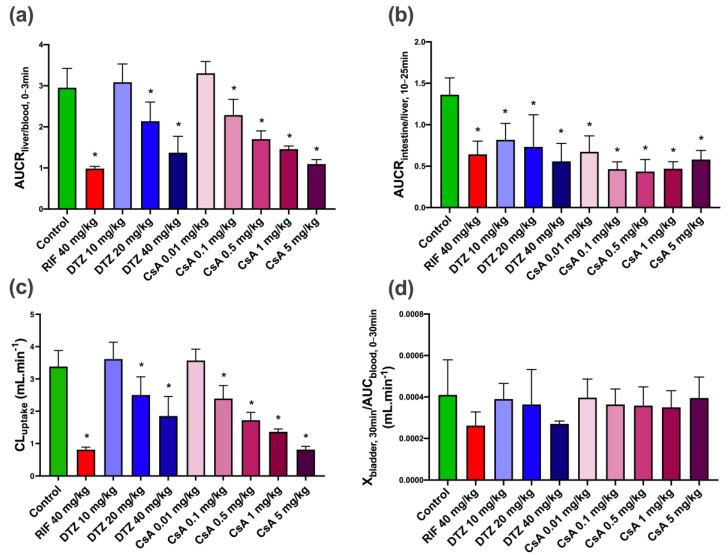
Mean descriptive pharmacokinetic parameters for control animals and animals treated by rifampicin (RIF), diltiazem (DTZ) and cyclosporin A (CsA). (**a**) AUCR_liver/blood,0–3 min_ (indicative of uptake of [^99m^Tc]mebrofenin from blood into liver), (**b**) AUCR_intestine/liver,10–25 min_ (indicative of excretion of [^99m^Tc]mebrofenin from the liver into the intestine), (**c**) CL_uptake_ values calculated from the integration plot analysis and (**d**) X_bladder,30 min_/AUC_blood,0–30 min_ (corresponding to CL_urine_). Data are means with error bars representing SD (*n* = 5 for control, RIF, DTZ 10 mg/kg, DTZ 20 mg/kg, DTZ 40 mg/kg, CsA 0.1 mg/kg groups and *n* = 6 for CsA 0.01 mg/kg, CsA 0.5 mg/kg, CsA 1 mg/kg, CsA 5 mg/kg groups). *, *p* < 0.05 compared to control, one-way ANOVA with a Bonferroni multiple-comparison test.

**Figure 6 pharmaceutics-12-00486-f006:**
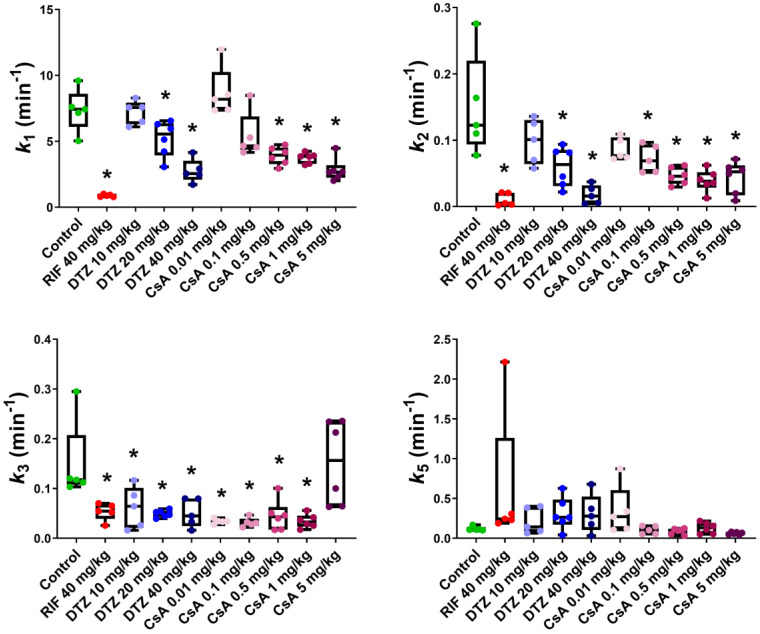
Outcome parameters for [^99m^Tc]mebrofenin hepatobiliary distribution obtained with the four-compartment pharmacokinetic model for control animals and animals treated by rifampicin (RIF), diltiazem (DTZ) and cyclosporin A (CsA). *k*_1_ represents the transfer of [^99m^Tc]mebrofenin from blood into the hepatocytes, *k*_2_ the backflux of [^99m^Tc]mebrofenin from the hepatocytes into blood, *k*_3_ the transfer from the hepatocytes into the intrahepatic bile ducts and *k*_5_ from the intrahepatic bile duct to the intestine (*n* = 5 for control, RIF, DTZ 10 mg/kg, DTZ 20 mg/kg, DTZ 40 mg/kg, CsA 0.1 mg/kg groups and *n* = 6 for CsA 0.01 mg/kg, CsA 0.5 mg/kg, CsA 1 mg/kg, CsA 5 mg/kg groups). *, *p* < 0.05 compared to control, one-way ANOVA with a Bonferroni multiple-comparison test.

**Figure 7 pharmaceutics-12-00486-f007:**
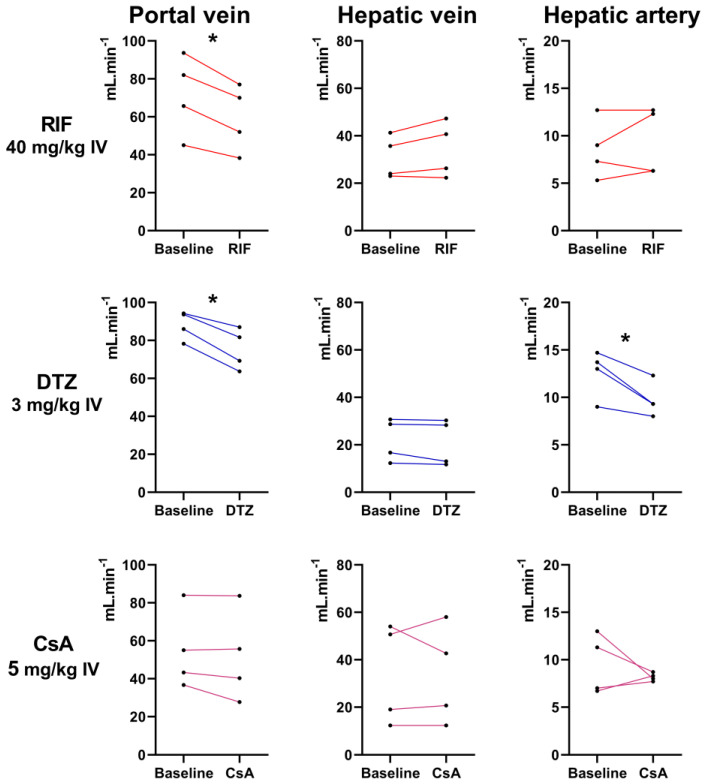
Blood flow (mL/min) of the portal vein, hepatic vein and hepatic artery measured with Doppler ultrasound in 4 rats before and after injection of rifampicin (RIF), diltiazem (DTZ) and cyclosporin A (CsA), respectively. *, *p* < 0.05, paired *t*-test.

**Table 1 pharmaceutics-12-00486-t001:** Descriptive pharmacokinetic parameters of [^99m^Tc]mebrofenin under each tested condition.

Group	AUC_0–30 min_ Blood ([cps/MBq].min)	AUC_0–30 min_ Liver ([cps/MBq].min)	X_max_ Liver (cps/MBq)	T_max_ Live (min)	AUC_0–30 min_ Intestine ([cps/MBq].min)
**Control** (*n* = 5)	51 ± 14	461 ± 109	26.5 ± 5.1	4.3 ± 1.7	424 ± 85
**RIF** (*n* = 5)	109 ± 12 *	385 ± 28	16.0 ± 0.9 *	9.6 ± 0.8 *	223 ± 54 *
**DTZ 10 mg/kg** (*n* = 5)	47 ± 4	520 ± 61	30.1 ± 3.2	4.3 ± 0.7	271 ± 30 *
**DTZ 20 mg/kg** (*n* = 5)	72 ± 23	562 ± 96	31.1 ± 5.8	5.4 ± 1.6	286 ± 102 *
**DTZ 40 mg/kg** (*n* = 5)	84 ± 30 *	543 ± 95	24.3 ± 3.0	6.4 ± 0.8	238 ± 76 *
**CsA 0.01 mg/kg** (*n* = 6)	71 ± 9	660 ± 61 *	34.6 ± 3.0 *	4.0 ± 0.6	327 ± 61
**CsA 0.1 mg/kg** (*n* = 5)	72 ± 11	601 ± 105 *	27.9 ± 4.0	5.7 ± 1.9	223 ± 26 *
**CsA 0.5 mg/kg** (*n* = 6)	97 ± 9 *	668 ± 48 *	28.9 ± 2.1	7.6 ± 2.1 *	239 ± 64 *
**CsA 1 mg/kg** (*n* = 6)	124 ± 9 *	695 ± 40 *	29.9 ± 2.5	7.4 ± 1.3 *	275 ± 38 *
**CsA 5 mg/kg** (*n* = 6)	188 ± 18 *	648 ± 66 *	27.6 ± 2.1	9.1 ± 1.3 *	314 ± 30 *

Data are given as mean ± SD (*n* = 5 for control, RIF, DTZ 10 mg/kg, DTZ 20 mg/kg, DTZ 40 mg/kg, CsA 0.1 mg/kg groups and *n* = 6 for CsA 0.01 mg/kg, CsA 0.5 mg/kg, CsA 1 mg/kg, CsA 5 mg/kg groups). Asterisk (*) indicates significant differences (*p* < 0.05) in animals treated by rifampicin (RIF), diltiazem (DTZ) or cyclosporin A (CsA) compared with the control group (one-way ANOVA with a Bonferroni multiple-comparison test).

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
