# Peer review of "Validation of Pharmacological Protocols for Targeted Inhibition of Canalicular MRP2 Activity in Hepatocytes Using [99mTc]mebrofenin Imaging in Rats"

_pharmaceutics, 2020, doi:10.3390/pharmaceutics12060486_

Round 1

Reviewer 1 Report

48 Figure 1. Hepatic transporters and rate constants from the four-compartment pharmacokinetic model.

OBS In ref 4 (Hernández Lozano, I.; Karch, R.; Bauer, M.; Blaickner, M.; Matsuda, A.; Wulkersdorfer, B.; Hacker, M.;

Zeitlinger, M.; Langer, O. Towards Improved Pharmacokinetic Models for the Analysis of Transporter-

Mediated Hepatic Disposition of Drug Molecules with Positron Emission Tomography. AAPS J 2019, 435 61)  are presented a four compartmental model and a three compartmental model .  In figure 1 of your paper is presented in fact  the three compartmental model.

Obs,  Maybe in figure 2 would be better to represent the standard deviations of the data also/

Compartmental kinetic modeling showed that the transfer rate constant of [99mTc]mebrofenin

275 from blood into hepatocytes k1 (Figure 1) was significantly decreased by RIF (k1 = 0.89 ± 0.09 min-1)

276 and, to a lesser extent, by the highest doses of DTZ (k1 = 2.76 ± 0.89 min-1) and CsA (k1 = 2.82 ± 0.86

277 min-1) as compared with the control group (k1 = 7.37 ± 1.62 min-1) (Figure 6). A dose-dependent

278 decrease in k1 was observed with increasing doses of DTZ and CsA, with no effect observed for 10

279 mg/kg DTZ and 0.01 mg/kg CsA. The backflux of [99mTc]mebrofenin from hepatocytes into blood k2

280 (Figure 1) was dose-dependently decreased by DTZ and CsA (Figure 6). k2 was not decreased by the

281 lowest dose of DTZ and CsA. The transfer of [99mTc]mebrofenin from the hepatocyte into the

282 intrahepatic bile ducts k3 (Figure 1), which reflected MRP2-mediated excretion, was significantly

Observation.  It is not sufficient to give the values of constants  k1, k2  and k3. It is necessary to validate , at least partially,  its.  It is necessary to compare  the  estimated concentrations curve with experimental data. A graphical presentation would be sufficient. 

289 Figure 6. Outcome parameters for [99mTc]mebrofenin hepatobiliary distribution obtained with the …

Some values in control  of  k2 , k3  and k4  are clearly outliers . It is necessary to make another analysis  after their elimination. After this to  compare the two sets of results.  In case of dramatically changes of the conclusions  ,  it is necessary a more in depth evaluation of data.  It is again necessary  in this context , to evaluate the reliability  of  individual sets of constants by a  comparison (visually)  the theoretical curves and experimental data.

Author Response

We thank the reviewers for their constructive comments. We addressed all points they have raised and changed the manuscript accordingly. Changes to the original manuscript are highlighted in yellow.

Reviewer 1

  1. 48 Figure 1. Hepatic transporters and rate constants from the four-compartment pharmacokinetic model.

OBS In ref 4 (Hernández Lozano, I.; Karch, R.; Bauer, M.; Blaickner, M.; Matsuda, A.; Wulkersdorfer, B.; Hacker, M.; Zeitlinger, M.; Langer, O. Towards Improved Pharmacokinetic Models for the Analysis of Transporter-Mediated Hepatic Disposition of Drug Molecules with Positron Emission Tomography. AAPS J 2019, 435 61)  are presented a four compartmental model and a three compartmental model.  In figure 1 of your paper is presented in fact  the three compartmental model.

We have indeed used a four-compartment model to describe the kinetics of [99mTc]mebrofenin. We have removed the rate constants from Figure 1 (lines 47-53) and added representation of the four-compartment model as supplementary data (Figure S1, lines 448-452).

  1. Obs,Maybe in figure 2 would be better to represent the standard deviations of the data

The standard deviations have been added to Figure 2 (lines 226-230, n = 4 for each measurement). We also added information regarding the method used to fit in vitrodata and estimate IC50, which was missing in the previous version of the manuscript (lines 134-136).

  1. Line 275 – 282 Compartmental kinetic modeling showed that the transfer rate constant of [99mTc]mebrofenin from blood into hepatocytes k1 (Figure 1) was significantly decreased by RIF (k1 = 0.89 ± 0.09 min-1) and, to a lesser extent, by the highest doses of DTZ (k1 = 2.76 ± 0.89 min-1) and CsA (k1 = 2.82 ± 0.86 min-1) as compared with the control group (k1 = 7.37 ± 1.62 min-1) (Figure 6). A dose-dependent decrease in k1 was observed with increasing doses of DTZ and CsA, with no effect observed for 10 mg/kg DTZ and 0.01 mg/kg CsA. The backflux of [99mTc]mebrofenin from hepatocytes into blood k2(Figure 1) was dose-dependently decreased by DTZ and CsA (Figure 6). k2 was not decreased by the lowest dose of DTZ and CsA. The transfer of [99mTc]mebrofenin from the hepatocyte into the intrahepatic bile ducts k3 (Figure 1), which reflected MRP2-mediated excretion, was significantly

Observation.  It is not sufficient to give the values of constants  k1, k2  and k3. It is necessary to validate, at least partially,  its.  It is necessary to compare  the  estimated concentrations curve with experimental data. A graphical presentation would be sufficient. 

The authors appreciate this comment and agree with the reviewer. We have now included a supplementary figure (Figure S3, lines 460-463) with the fits of the model for the liver and intestine time-activity curves of one representative subject for each of the study groups.

In addition, we have included a sentence in the results section regarding the suitability of the chosen pharmacokinetic model to represent the observed data (section 3.3., lines 299-301): “We used a previously developed four-compartment PK model which provided good fits (Figure S3) for both the liver and the intestine ROI. Precision of the obtained parameters was acceptable (low percentage coefficient of variation, %CV) in almost all the analyzed subjects (Table S1).”

Moreover, we have included a new table (Table S1, lines 464-473) summarizing the mean obtained parameter values and the percentage coefficient of variation of the parameters (%CV), since this allows to see that the parameter precision is, in general, good, and we can rely on the obtained model parameters.

  1. 289 Figure 6Outcome parameters for [99mTc]mebrofenin hepatobiliary distribution obtained with the …

Some values in control  of  k2 , k3  and k4  are clearly outliers . It is necessary to make another analysis  after their elimination. After this to  compare the two sets of results.  In case of dramatically changes of the conclusions  ,  it is necessary a more in depth evaluation of data.  It is again necessary  in this context , to evaluate the reliability  of  individual sets of constants by a  comparison (visually)  the theoretical curves and experimental data.

As suggested by the reviewer, we performed the statistical analysis for k2, k3, and k5after removing the outliers in the control group (or in the rifampicin group for k5). Statistically significant differences remained the same for k2and k5as compared to the original analysis. However, when the outlier value for k3in the control group was removed, some changes occur in the significance levels. This is now clearly mentioned in the results (section 3.3, lines 312-316): “Statistically significant differences remained the same for k2and k5after excluding outlier data. However, the significance is lost in some of the groups when removing the outlier value for k3(0.29 min-1) in the control group. In this situation, only the 20 mg/kg DTZ, and all the CsA groups except the 5 mg/kg group show a significant decrease in the k3value as compared to the control group.

Since the new analysis does not dramatically change the observed results, we prefer leaving Figure 6 as it is. In addition, the formerly obtained results (without outlier exclusion) agreed with the parameters obtained with the non-compartmental pharmacokinetic analysis (Figure 5b), which serves as confirmation of the reliability of individual sets of constants. Moreover, as mentioned above, a table (Table S1, lines 464-473) including mean parameter values and the percent coefficient of variation for each individual set of rate constants was included to show robustness of the model and precision of the estimated pharmacokinetic parameters.

Reviewer 2 Report

This manuscript demonstrated that low-dose CsA selectively decreased the transfer of mebrofenin from the liver to the bile. Targeted pharmacological inhibition of hepatic MRP2 activity can be achieved without impacting OATP activity and liver blood flow in rats. The studies provide important findings for pre-clinical studies, or potentially for the clinical care.

Comments:

  1. Please add the number of rats in each group in the legends for Figures 4-6 and Table 1.

Author Response

We thank the reviewers for their constructive comments. We addressed all points they have raised and changed the manuscript accordingly. Changes to the original manuscript are highlighted in yellow.

Comments:

  1. Please add the number of rats in each group in the legends for Figures 4-6 and Table 1.

The information was added lines 253-255 (figure 4), lines 260-262 (table 1), lines 286-287 (figure 5) and lines 326-327 (figure 6) : “(n = 5 for control, RIF, DTZ 10 mg/kg, DTZ 20 mg/kg, DTZ 40 mg/kg, CsA 0.1 mg/kg groups and n = 6 for CsA 0.01 mg/kg, CsA 0.5 mg/kg, CsA 1 mg/kg, CsA 5 mg/kg groups)”. We also added this information to the new Table S1 (lines 466-468).

Reviewer 3 Report

In the manuscript entitled “Validation of pharmacological protocols for targeted inhibition of canalicular MRP2 activity in hepatocytes using [99mTc]mebrofenin imaging in rats,” the authors tried different doses of known MRP2 inhibitors, i.e., diltiazem (DTZ) and cyclosporin A (CSA), to determine the doses needed for targeted pharmacological inhibition of hepatic MRP2 activity in vivo. While the goal of the study seems straightforward, a number of major concerns are noted especially in the scientific rationales for study design as well as in the interpretation of data.

Major comments:

  1. Introduction:
    1. Please include information on currently available knowledge about DTZ/CSA effects on other transporters involved in mebrofenin disposition (i.e., MRP3, OATP1B1/1B3).
    2. It appears that p-glycoprotein is not involved in mebrofenin disposition, yet CSA inhibition of P-gp was mentioned. If possible, it would help readers that the authors focus on the transporters for mebrofenin.
    3. It is stated that DTZ and CSA are known MRP2 inhibitors. Please add references. Were IC50 (or Ki) values of DTZ/CSA reported in previous studies for MRP2 inhibition? If yes, the need for in vitro inhibition analysis (Fig 2) seems unclear.
  2. DTZ/CSA dose: Please provide rationales for choosing the doses for MRP2 inhibition. For example, would the doses lead to plasma or liver concentrations high enough for MRP2 inhibition? This can be estimated based on previously reported pharmacokinetic parameters including volume of distribution.
  3. DTZ route of administration: While other inhibitors were administered intravenously (IV), DTZ was given subcutaneously (SC). Please provide a rationale for choosing different routes of administrations for DTZ vs CSA.
  4. DTZ Doppler ultrasound study: It is unclear why a different route of administration (IV) was used for Doppler study than the scintigraphy study (SC). The DTZ disposition after IV study does not reflect what happens after SC, and thus the Doppler data may not be translated to interpret the scintigraphy results.
  5. Pharmacokinetic modeling:
    1. It appears that based on scintigraphy readout from liver, intestine, and heart, the authors estimated four rate constants including k3 (hepatocytes to intrahepatic bile ducts) and k5 (intrahepatic bile ducts to intestine). Not enough information is given to justify the need for two parameters to describe biliary drug excretion (e.g., the processes are regulated independently?). Also, the robustness of k3 and k5 estimation is unclear because apparently there was no readout for “intrahepatic bile ducts”.
    2. Significant decrease in k2 by rifampin is noted. Has rifampin inhibition of MRP2 been reported previously? Such information, if available, may support the robustness of modeling process.
  6. The authors claim that DTZ and CSA, at the lowest doses, inhibit MRP2 in vivo without affecting OATP1B1/1B3 or MRP3. Considering that [99mTc]mebrofenin disposition has been studied in mrp2-null mice (reference 40), the authors should compare the data in mrp2-null mice against the results from the lowest doses of DTZ or CSA, and discuss any discrepancy if noted.
  7. Line 334: The statement that “we hypothesized that CSA would be more potent in inhibiting MRP2 than OATPs” is missing a scientific rationale.
  8. Line 369, The statement that “OATP inhibition increased the plasma exposure increased the plasma exposure… (Table 1)” is not supported by the data. Rifampin increased the plasma exposure (Table 1), but this may result from inhibition of OATP as well as MRP2.

Minor comments:

  1. Fig S1: Graph legend is missing.

Author Response

We thank the reviewers for their constructive comments. We addressed all points they have raised and changed the manuscript accordingly. Changes to the original manuscript are highlighted in yellow.

Major comments:

  1. Introduction

a) Please include information on currently available knowledge about DTZ/CSA effects on other transporters involved in mebrofenin disposition (i.e., MRP3, OATP1B1/1B3).

We now provide available information regarding the potential of tested inhibitors (rifampicin, diltiazem and cyclosporin) against transporters involved in mebrofenin disposition in the introduction. Please see, lines 92-97: “There are limited data regarding the impact of transporter inhibitors on the liver kinetics of [99mTc]mebrofeninin vivo. RIF is a potent in vitroinhibitor of OATP [28], MRP2 [8] and MRP3 [29]. In vivoexperiments in rats and mice showed that RIF inhibited both the liver uptake and canalicular efflux of [99mTc]mebrofenin to a similar extent [21,30]. Diltiazem (DTZ) is a potent in vitroinhibitor of MRP2 [8], but not of OATP1B1 and OATP1B3 [28]. Cyclosporin (CsA) is a potent in vitroinhibitor of MRP2 [8] and MRP3 [31] but is also known to inhibit OATP [28].

b) It appears that p-glycoprotein is not involved in mebrofenin disposition, yet CSA inhibition of P-gp was mentioned. If possible, it would help readers that the authors focus on the transporters for mebrofenin.

We agree with this comment and removed any mention of P-gp inhibition in the introduction(section 1, lines 87-89): “Examples are the combination of the P-glycoprotein (P-gp) imaging probes [11C]verapamil or [11C]metoclopramide with the P-gp inhibitors tariquidar [28–30] or cyclosporin A (CsA) [31] to study P-gp activity at the blood-brain barrier (BBB), oris the combination of OATP substrate radiotracers (such as [11C]dehydropravastatin) with the OATP inhibitor rifampicin (RIF) to study OATP activity in the liver [27].

c) It is stated that DTZ and CSA are known MRP2 inhibitors. Please add references. Were IC50 (or Ki) values of DTZ/CSA reported in previous studies for MRP2 inhibition? If yes, the need for in vitro inhibition analysis (Fig 2) seems unclear.

Corresponding reference has been added to the introduction lines 96-97 (Matsson 2009, reference 8). This screening paper did not compare the potency of compounds to inhibit MRP2 (IC50). We think it is important to show our in vitrodata that enabled estimation of IC50and allow for direct comparison of the MRP2 inhibitory potency of RIF, CsA and DTZ in the same conditions. A sentence has been added to explain the added value of our in vitro assay (line 119) : “We used an in vitro assay to compare the MRP2-inhibitory potency of RIF, DTZ and CsA".

  1. and 3. DTZ/CSA dose: Please provide rationales for choosing the doses for MRP2 inhibition. For example, would the doses lead to plasma or liver concentrations high enough for MRP2 inhibition? This can be estimated based on previously reported pharmacokinetic parameters including volume of distribution.

DTZ route of administration: While other inhibitors were administered intravenously (IV), DTZ was given subcutaneously (SC). Please provide a rationale for choosing different routes of administrations for DTZ vs CSA.

Doses and routes of administration of tested inhibitors have been selected upon available PK and tolerance data, in a clinical perspective when possible. Detailed rationale for selected dosing is now provided in section 2.4. Please see lines 144-149: “The PK of subcutaneously (s.c.) injected DTZ (20 mg/kg) has been reported in rats with plasma concentration > 10 µg/mL (25 µM) obtained 60 min after injection [34], consistent with the potency of DTZ to inhibit MRP2, assessed in vitroin the present study. A dose range of DTZ (10, 20 or 40 mg/kg s.c.) was therefore administered 60 minutes before [99mTc]mebrofenin. CsA was injected i.v. 5 minutes before [99mTc]mebrofenin, starting from a high clinical dose which was further decreased (5, 1, 0.5, 0.1, 0.01 mg/kg, i.v.).

  1. DTZ Doppler ultrasound study: It is unclear why a different route of administration (IV) was used for Doppler study than the scintigraphy study (SC). The DTZ disposition after IV study does not reflect what happens after SC, and thus the Doppler data may not be translated to interpret the scintigraphy results.

We now provide explanation for the different route of administration used for DTZ in the Doppler study compared with the scintigraphy study (section 2.7, lines 207-210) : “Flow in the three vessels was measured before (= baseline) and 5 min after injection of maximal dose of the tested inhibitors to allow for paired measurements, without moving the US probe, over a limited time period. Therefore, all three inhibitors, including DTZ, had to be injected i.v. to enable rapid detection of drug-induced changes in blood flow within a single session. RIF was administered 40 mg/kg i.v. (n = 4) and CsA was administered 5 mg/kg i.v. (n = 4). For DTZ, it was not technically possible to match the conditions used for [99mTc]mebrofenin scintigraphy and wait for 60 min after s.c. administration. Intravenous injection of 40 mg/kg dose of DTZ was lethal (n = 3). DTZ was therefore administered at the maximal i.v. dose reported in the literature in rats (3 mg/kg i.v.; n = 4) [39].

This is also highlighted in the discussion to avoid any overinterpretation of the results (section 4, lines 377-380): “For technical reasons, it was not possible to test the impact of DTZ 40 mg/kg s.c. on liver perfusion. However, DTZ (3 mg/kg i.v.) was shown to induce a modest but significant decrease in blood flow in both the portal vein and hepatic artery (Figure 7).”

We also clarified the dose of DTZ used for the Doppler study in the Abstract. Please see line 31: “High dose of DTZ (i.v., 3mg/kg) but not CsA (i.v., 5mg/kg)

  1. Pharmacokinetic modeling:a) It appears that based on scintigraphy readout from liver, intestine, and heart, the authors estimated four rate constants including k3 (hepatocytes to intrahepatic bile ducts) and k5 (intrahepatic bile ducts to intestine). Not enough information is given to justify the need for two parameters to describe biliary drug excretion (e.g., the processes are regulated independently?). Also, the robustness of k3 and k5 estimation is unclear because apparently there was no readout for “intrahepatic bile ducts”.

The rationale behind separating k3and k5is that the processes they represent (transport from hepatocytes to intrahepatic bile duct and transport from intrahepatic to extrahepatic bile ducts, respectively) are two independent processes. While the transfer from hepatocytes into the intrahepatic bile ducts is regulated by transporters (Figure 1, Figure S1), the transport from the intrahepatic bile ducts to the extrahepatic bile ducts and intestines is not (it represents the bile flow). This model was previously developed by Orntoft et al. (Orntoft 2017, reference 37) and afterwards modified by Hernández Lozano et al. (Hernández Lozano 2019, reference 36) to describe the hepatic disposition of different radiotracers in humans and the robustness of the k3and k5estimates was shown to be high although no imaging data were available for the “intrahepatic bile ducts”. As pointed out by the reviewer, there is no readout for intrahepatic bile ducts in the present study, but based on liver physiology and assuming similarities between human and rat hepatic anatomy and physiology, one can estimate the expected concentration and volumes in the individual depicted compartments.

Moreover, to assess the reliability/robustness of the model to represent the observed data and the parameter precision, the percentage coefficient of variation (%CV) was calculated for each individual estimated parameter. Table S1 (lines 464-473) was included in the manuscript showing the average parameter values in addition to the %CV range in each study group. A comparison of the observed against the predicted radioactivity curves (new Figure S3, lines lines 460-463) also shows that the model can accurately represent the observed data.

We have included a few additional sentences in the text in order to describe in further detail these aspects of the pharmacokinetic model (section 2.6, lines 183-188): « The processes defined by k1,k2, and k3may involve active transport through the basolateral or canalicular membranes of hepatocytes. The liver volume of interest defined in the model consists of well mixed compartments of blood, hepatocytes and intrahepatic bile ducts. The additional compartment represents the extrahepatic bile ducts and intestines. Compartment volumes are estimated by hepatic anatomical characteristics and physiological processes as previously described [36-37].In addition… »

Results, section 3.3, lines 299-301: « We used a previously developed four-compartment PK model which provided good fits (Figure S3) for both the liver and the intestine ROI. Precision of the obtained parameters was acceptable (low percentage coefficient of variation, %CV) in almost all the analyzed subjects (Table S1). »

b) Significant decrease in k2 by rifampin is noted. Has rifampin inhibition of MRP2 been reported previously? Such information, if available, may support the robustness of modeling process.

Inhibition of MRP2 by rifampicin has already been reported in reference 8 (Matsson et al. 2009, line 93). The authors assume that the reviewer refers to an inhibition of MRP3 as an explanation for the reduction of k2in the rifampicin group. We now provide information supporting the inhibition of MRP3 by rifampicin (Te Brake 2016, reference 29) in the introduction (line 93) and in the discussion (line 398).

  1. The authors claim that DTZ and CSA, at the lowest doses, inhibit MRP2 in vivo without affecting OATP1B1/1B3 or MRP3. Considering that [99mTc]mebrofenin disposition has been studied in mrp2-null mice (reference 40), the authors should compare the data in mrp2-null mice against the results from the lowest doses of DTZ or CSA, and discuss any discrepancy if noted.

It is an interesting point to add to the discussion and the authors are grateful to the reviewer for this relevant comment. Please see lines 404-405 (“This observation is consistent with the kinetics of [99mTc]mebrofenin obtained in MRP2-deficient mice [30].”) and 411-415 (“It is noteworthy that the baseline biliary excretion of [99mTc]mebrofenin reported in mice was much higher (calculated AUCRbile/liver~28) [30]. The impact of MRP2-deficiency was consistently much more pronounced, with a ~10-fold decrease in AUCRbile/liver[30]. This observation suggests species difference in the importance of MRP2 in controlling the biliary excretion of [99mTc]mebrofenin between rats and mice.”).

  1. Line 334: The statement that “we hypothesized that CSA would be more potent in inhibiting MRP2 than OATPs” is missing a scientific rationale.

We rephrased this sentence in order to better reflects our work hypothesis. Please see lines 367-368: “We hypothesized that CsA may show substantially different potency to inhibit MRP2 versusOATP in vivo[21,28].

  1. Line 369, The statement that “OATP inhibition increased the plasma exposure increased the plasma exposure… (Table 1)” is not supported by the data. Rifampin increased the plasma exposure (Table 1), but this may result from inhibition of OATP as well as MRP2.

We took this important comment into account and removed this sentence. Please see lines 403-404.

Minor comments:

  1. Fig S1: Graph legend is missing.

Figure S1 (now figure S2) now has its legend (lines 455-458).

Round 2

Reviewer 1 Report

authors had made significant corrections,  the paper is more clear and results more reliable interpreted

Reviewer 3 Report

All my concerns were addressed.